# Lipids and Lipoproteins in Health and Disease: Focus on Targeting Atherosclerosis

**DOI:** 10.3390/biomedicines9080985

**Published:** 2021-08-09

**Authors:** Chih-Kuo Lee, Che-Wei Liao, Shih-Wei Meng, Wei-Kai Wu, Jiun-Yang Chiang, Ming-Shiang Wu

**Affiliations:** 1College of Medicine, National Taiwan University, Taipei 100, Taiwan; keitheva2009@gmail.com (C.-K.L.); yo.ahliao@gmail.com (C.-W.L.); volcano6369@gmail.com (S.-W.M.); weikaiwu0115@gmail.com (W.-K.W.); 2Department of Internal Medicine, National Taiwan University Hospital Hsin-Chu Branch, Hsin-Chu 300, Taiwan; 3Graduate Institute of Clinical Medicine, College of Medicine, National Taiwan University, Taipei 100, Taiwan; 4Department of Internal Medicine, National Taiwan University Cancer Center, Taipei 106, Taiwan; 5Division of Cardiology, Department of Internal Medicine, National Taiwan University Hospital Hsin-Chu Branch, Hsin-Chu 300, Taiwan; 6Division of Gastroenterology and Hepatology, Department of Internal Medicine, National Taiwan University Hospital, National Taiwan University College of Medicine, Taipei 100, Taiwan; 7Division of Cardiology, Department of Internal Medicine and Cardiovascular Center, National Taiwan University Hospital, Taipei 100, Taiwan

**Keywords:** low-density lipoprotein, high-density lipoprotein, triglyceride, apolipoprotein, lipoprotein(a)

## Abstract

Despite advances in pharmacotherapy, intervention devices and techniques, residual cardiovascular risks still cause a large burden on public health. Whilst most guidelines encourage achieving target levels of specific lipids and lipoproteins to reduce these risks, increasing evidence has shown that molecular modification of these lipoproteins also has a critical impact on their atherogenicity. Modification of low-density lipoprotein (LDL) by oxidation, glycation, peroxidation, apolipoprotein C-III adhesion, and the small dense subtype largely augment its atherogenicity. Post-translational modification by oxidation, carbamylation, glycation, and imbalance of molecular components can reduce the capacity of high-density lipoprotein (HDL) for reverse cholesterol transport. Elevated levels of triglycerides (TGs), apolipoprotein C-III and lipoprotein(a), and a decreased level of apolipoprotein A-I are closely associated with atherosclerotic cardiovascular disease. Pharmacotherapies aimed at reducing TGs, lipoprotein(a), and apolipoprotein C-III, and enhancing apolipoprotein A-1 are undergoing trials, and promising preliminary results have been reported. In this review, we aim to update the evidence on modifications of major lipid and lipoprotein components, including LDL, HDL, TG, apolipoprotein, and lipoprotein(a). We also discuss examples of translating findings from basic research to potential therapeutic targets for drug development.

## 1. Introduction

During the past decades, the risk of atherosclerotic cardiovascular disease (ASCVD) and mortality has been much reduced due to advances in pharmacotherapy, intervention devices, and techniques. [1,2] ASCVD risk is significantly reduced by controlling blood low-density lipoprotein cholesterol (LDL-C) level [3]. Statin is the drug of choice to treat hypercholesterolemia. Non-statin medication including PCSK9 inhibitors and ezetimibe would further reduce LDL-C level while added to a statin or act as statin alternatives. Bempedoic acid is a newly approved effective non-statin LDL-C lowering agent [4]. Bempedoic acid has been associated with increased incidence of hyperuricemia, gout, and elevated serum creatinine level. On-going trials will clarify its long-term effect on cardiovascular outcomes [5].

However, the risk of ASCVD has not been eliminated. In the CANTOS trial, patients with a history of myocardial infarction (MI) had a 20% 5-year rate of recurrent major cardiovascular events (MACEs) despite statin treatment [6]. These residual risks can be caused by many factors, and methods to modify these factors have been proposed in contemporary guidelines. For example, measuring the lipoprotein(a) (Lp(a)) level should be considered among high-risk patients for a more precise reclassification and identification [3,7]. Lipid and lipoprotein metabolism disorders remain an unsolved problem. Whilst most guidelines encourage achieving target levels of specific lipoproteins to reduce the risk of ASCVD, increasing evidence has shown that molecular modification of these lipoproteins also has a critical impact on their atherogenecity and may contribute to residual ASCVD risk (Figure 1). For example, native low-density lipoproteins (LDLs) are much less atherogenic than those that have been structurally modified, such as by oxidation [8]. Apolipoproteins also play important roles in modulating lipid homeostasis and may alter the functions of different lipoproteins. In this review, we aim to update the evidence on modifications of major lipid components, including LDL, high-density lipoprotein (HDL), triglycerides (TGs), apolipoprotein, and Lp(a). We also discuss examples of translating findings from basic research to potential therapeutic targets for drug development.

## 2. Low-Density Lipoprotein Cholesterol

An increased serum LDL level is a well-known major risk factor for atherosclerotic disease. LDL is the major cholesterol carrier in plasma, consisting mainly of cholesterol ester (29%), phospholipids (28%), protein (21%), free cholesterol (11%), and TGs (9%). The only static protein component of LDL is apolipoprotein B (apoB) [10]. Mediated by apoB, LDL binds to and is internalized by LDL receptors (LDLRs) expressed on endothelial cells, macrophages, monocytes, and smooth muscle cells [11]. However, since LDLRs are downregulated as the intracellular cholesterol level increases, the uptake of native LDL via LDLRs does not consequently cause a significant increase in total cholesterol. In contrast, macrophage uptake modifies LDL through scavenger receptors, which are not downregulated as the cholesterol level increases, leading to substantial cholesterol accumulation and subsequent foam cell formation [12]. Therefore, modified LDL is assumed to be more atherogenic than native LDL.

LDL can be modified in several ways and mainly by oxidation. Oxidation of LDL is generally thought to take place in the subendothelial space but not in the circulation, as the plasma is rich in antioxidants. LDL can be oxidized non-enzymatically by metal ions or enzymatically via various mechanisms in the arterial wall [8].

The oxidation of LDL results in a broad spectrum of oxidized LDL, from minimally oxidized LDL (mmLDL) to fully oxidized LDL (oxLDL) [13]. mmLDL represents the initial stage of LDL oxidation, with no or little change in apoB, and retaining the affinity to LDLRs [14]. mmLDL increases chemokines and cytokines and enhances interactions between monocytes and endothelial and inflammatory cell recruitment. Subsequently, LDL is further oxidized with more extensive modification of its proteins, losing recognition by LDLRs and shifting affinity to a variety of scavenger receptors, including class A (SR-AI and SR-AII), class B (CD36), class D (CD68), class E (LOX-1), and class G (SR-PSOX) [8,12,15,16,17,18].

In vitro, LDL can be oxidized by iron and copper [19], and this can be blocked by metal chelators [20]. Although elevated iron and copper levels have been detected in human carotid atherosclerotic lesions [21], the relationship between serum iron and copper levels with atherosclerosis is unclear. Not only do patients with hemochromatosis and Wilson disease not have a higher risk of atherosclerosis [22], epidemiological studies have also reported inconclusive results with regards to the relationship between plasma iron level and atherosclerosis [23]. Consequently, further studies are needed to elucidate the actual mechanism of transition ions in LDL oxidation in vivo.

Lipoxygenase is an intracellular enzyme that directly oxygenates poly-unsaturated fatty acids [24]. In vitro, 15-lipoxygenase has been shown to oxidizes LDL via direct and non-direct reactions [25]. In human atherosclerotic lesions and macrophage-rich areas of fatty streaks, elevated co-localization of oxidized LDL epitope, 15-lipoxygenase mRNA and protein, and abundant acetyl LDL receptor mRNA have been shown [26]. Further studies have reported compelling results, indicating that lipoxygenase may not only be involved in atherogenesis but also exert an anti-inflammatory effect [27].

Myeloperoxidase (MPO) is abundant in the azurophilic granules of leukocytes (neutrophils, monocytes), and it has been linked to inflammation and oxidative stress. Local release from resident macrophages and transcytosis of intra-luminally produced MPO by activated leukocytes has been shown to contribute to MPO production in the vascular wall [28,29]. Enzymatically active MPO has been shown in human vascular atherosclerotic lesions, indicating the involvement of MPO in atherosclerosis [30].

Glycation to LDL occurs non-enzymatically to the major amino acid of apoB, lysine, resulting in 2%–17% of LDL-lysine glycation [31]. In vivo, free radicals generated from glucose and Amadori products by glycation show the concurrence of glycation and oxidation, and glycated LDL is more susceptible to oxidation [32]. LDL glycation inevitably results in a certain level of LDL oxidation [33].

Nitric oxide (NO) plays an important role in vascular physiology. Despite the antioxidant property of NO, the concurrent formation of NO and superoxide anions potentiate peroxynitrite production and lipid peroxidation [34]. Part of the cell-mediated LDL oxidation in the arterial wall can be attributed to superoxide anions, peroxynitrite, or other reactive nitrogen intermediates produced by endothelial cells, smooth muscle cells, or macrophages [35].

LDL is a type of lipoprotein with heterogenous proteomic and lipidomic profiles and pathophysiological activity. Small dense LDL is a specific subclass of LDL considered to be more prominent in atherogenesis. In the placebo group of a large statin clinical trial, small dense LDL concentration but not large LDL was associated with higher cardiovascular risk [36]. Small dense LDL has several characteristics distinct from large LDL, including being enriched with apolipoprotein C-III (apoC-III) and glycated apoB [37], and having unsaturated cholesterol esters markedly susceptible to hydroperoxide formation under oxidative stress [38], which may be associated with their pathogenicity. In addition, compared with large LDL, small dense LDL has a longer residence time in the circulation, which is considered to increase the risk of atherosclerosis. The longer residence time may be due to more particle oxidation, modification, reduction in size, and increased arterial wall uptake [39].

Activated by oxLDL, endothelial cells can up-regulate adhesion molecules and chemokines, triggering the recruitment of monocytes, typically Ly6C^hi^ monocytes into the arterial wall [40]. These recruited monocytes differentiate into macrophages and further oxidize LDL. The oxLDL is then recognized and internalized by various scavenger receptors, turning the macrophages into cholesterol-laden foam cells [41]. By engaging pattern recognition receptors, such as toll-like receptors (TLRs), several DAMPs (damage-associated molecular patterns, notably oxidation-specific epitopes, and cholesterol crystals) generated by modification of retained LDL trigger the expressions of pro-inflammatory and pro-thrombotic genes in macrophages. This also boosts the recruitment of inflammatory cells, including monocyte-macrophages, neutrophils, lymphocytes, and dendritic cells [42]. Specifically, the recognition of oxLDL by the combination of TLR4-TLR6 and CD36 can promote the NFκB-dependent expression of chemokines, leading to further recruitment of macrophages [43]. ApoB is covalently modified during oxidation and apoB derived LDL-associated antigens are the most frequently studied putative T cell antigens. oxLDL-specific IgG levels have been shown to be well correlated with atherosclerosis progression and regression in animal models, however, the findings have been inconsistent in human studies. Antibody-dependent complement activation has also been shown in human atherosclerotic lesions [44]. The persistent presence of lipid-derived DAMPs, inflammatory cytokines, and the recruitment of phagocytes has been shown to sustain inflammatory responses and facilitates cross-talk with other arterial cells such as mast cells, contributing to plaque formation [45].

## 3. High-Density Lipoprotein Cholesterol

The concept of high-density lipoprotein cholesterol (HDL-C) as a “good cholesterol” can be traced to the Framingham Heart Study [46]. Subsequently, numerous epidemiological studies have suggested an inverse relationship between HDL-C level and the risk of ASCVD [47]. However, later efforts, including interventional studies focusing on niacin, fibrates, and cholesteryl ester transfer protein (CETP) inhibitors to increase HDL-C level, as well as meta-analyses, have failed to demonstrate a protective effect of a high HDL-C level against ASCVD [48,49,50,51]. These findings, along with several Mendelian randomization analyses, have prompted research that is focused on the functional properties rather than the quantity of HDL-C.

HDL is composed of a central hydrophobic non-polar lipid core, consisting of primarily triacylglycerols and cholesterol esters. HDL-C particles are highly heterogeneous, ranging in size from 5 to 17 nm and density from 1.063–1.210 kg/L, within human plasma [52]. Through lipidomic and proteomic techniques, more than 200 additional lipids and 85 proteins have been identified in HDL particles. Moreover, multiple enzymes and even genetic material in the form of micro RNAs have also been identified. These variations result in different sizes, densities, and functions of HDL. Based on the size and density, HDL is commonly classified into five broad subclasses, ranging from the largest and least dense (termed HDL2b) to the smallest and most dense (termed HDL3b) [53].

HDL proteins are composed of apolipoproteins (e.g. apoE, apoCs, apoA-IV, apoA-V, apoJ, apoF, apoM, apoL1), enzymes, lipid transfer proteins, acute-phase response proteins (including serum amyloid A [SAA]), complement components, proteinase inhibitors, and other protein components. Apolipoprotein A-I (apoA-I) is the major structural protein component of HDL and is primarily synthesized in the liver (~80%) and intestine (~20%), and then secreted in a lipid-free state. The enzymes carried by HDL may include those for lipid metabolism (e.g. lecithin-cholesterol acyltransferase [LCAT] and phospholipid transfer protein [PLTP]), and for antioxidation (e.g. paraoxonase 1 [PON1], platelet-activating factor-acetyl hydrolase [PAF-AH] and glutathione selenoperoxidase [GSPx]) [53].

The most well-known pathway for the anti-atherogenic nature of HDL is the ability to promote cholesterol efflux from cells through reverse cholesterol transport (RCT), which transports excess cholesterol from peripheral tissues to the liver for excretion. In addition, due to the numerous proteins carried by HDL as mentioned above, it also exhibits anti-oxidative, anti-thrombotic, and anti-inflammatory properties in normal physiology [54].

## 4. Dysfunctional HDL

As mentioned earlier, HDL-C is a highly heterogeneous particle with various sizes and additional apoproteins and enzymes. Hence, many alterations and conditions may cause dysfunctional HDL. In brief, compositional changes, post-translational modifications of proteins, or alterations in lipids and other cargo molecules, may result in functional differences [55,56,57,58].

The most commonly reported post-translational modifications include oxidation, carbamylation, and glycation [56]. HDL oxidation takes place predominantly in inflammatory conditions and frequently involves MPO. Excessive MPO expression has been observed during the progression of atherosclerotic plaques, and with a maximum expression just before plaque rupture. Carbamylation, a non-enzymatic and irreversible post-translational modification, is caused by interactions between isocyanic acid and various amino groups of proteins. Carbamylation also occurs in an inflammatory environment as a result of MPO-dependent cyanate formation. Glycation occurs after lipids or proteins are exposed to sugar. In chronic hyperglycemia (i.e. poorly controlled diabetes), nonenzymatic glycation may occur and lead to alterations in HDL composition (lipids, apoproteins, and enzymes) and functionality [59].

Modification of HDL can affect the lipidome and proteome then finally alter the function of HDL. Several pathways have been identified such as SAA, LCAT, CETP, apolipoprotein M, sphingosine-1-phosphate (S1P), paraoxonase 1, and MPO [56]. Many inflammatory processes can cause alteration in these pathways and modify HDL, which eventually losses its atheroprotective effect.

SAA is a family of apolipoproteins that are primarily synthesized in the liver. It is one of the main proteins of the acute phase response and plays a pivotal role in innate immunity [60,61]. The level of SAA has been shown to be higher during acute and chronic inflammation, which eventually causes compositional changes of HDL and the loss of its anti-inflammatory and atheroprotective potential.

LCAT esterifies free cholesterol and phosphatidylcholine into cholesteryl esters, and it is an essential enzyme for lipoprotein metabolism. Various acute and chronic inflammatory conditions can decrease LCAT activity, and subsequently, change the composition and function of HDL. This dysfunctional HDL also loses its atheroprotective potential.

CETP mediates the transfer of cholesterol esters from HDL to apoB-containing lipoproteins in exchange for TGs. Studies have shown a decreased level and activity of CETP in patients with sepsis, cardiac surgery, and rheumatoid arthritis. This decrease in CETP may cause remodeling of HDL and may increase the risk of cardiovascular mortality.

Apolipoprotein M (apoM) is a plasma protein of the apolipoprotein family expressed in the liver and kidneys. As mentioned earlier, apoM is an additional apolipoprotein carried by HDL. Previous studies have shown a decrease in the apoM level in inflammatory conditions, including sepsis, diabetes, and autoimmune disease, which in turn decrease the anti-inflammatory effects of HDL.

S1P is a lysosphingolipid found in association with small and dense HDL particles. HDL-bound S1P is important for endothelial survival, angiogenesis, migration, NO production, and inhibition of inflammatory responses [62]. The level of S1P has been found to be lower in patients with coronary artery disease and MI. Similar results have also been shown in patients with diabetes mellitus, chronic kidney disease, and atherosclerosis. 

PONs are a family of enzymes with antioxidative properties in mammals. They are composed of three different enzymes. HDL-associated PON1 possesses anti-atherogenic properties, protects LDL from oxidative modification, and promotes cholesterol efflux [63]. Many studies focusing on patients with obesity, diabetes mellitus, various autoimmune diseases, and infection have demonstrated decreased PON levels or activity, resulting in a possible decline in the anti-oxidative effect of HDL.

From a clinical perspective, many diseases have been correlated with dysfunctional HDL irrespective of its plasma level. For example, patients with acute coronary syndrome or stable coronary artery disease, have still be shown to exhibit reduced HDL-mediated cellular cholesterol efflux compared with healthy subjects even when matched by HDL-C level [64]. Alteration in the HDL function in such patients may be due to combinations of the abovementioned pathways. Other examples have also been reported, such as in patients with type 2 diabetes, in whom dysfunctional HDL may be caused by advanced glycation end products and increased oxidative stress with inflammation that reduces their ability to promote cholesterol efflux [59] or anti-inflammatory capacity [65]. In addition, HDL efflux capacity in dialytic patients has been shown to be dramatically reduced compared to matched subjects with normal kidney function, which could not be improved with statins [66]. In conclusion, current evidence supports that dysfunctional HDL, rather than HDL level, plays a more significant role in atherosclerosis. Further studies on the causality of the possible mechanisms of HDL modifications are needed.

## 5. Triglycerides

Hypertriglyceridemia is a prevalent condition observed in daily medical care. According to previous literature, its prevalence in the adult population is approximately 10% [67,68,69]. Moreover, the increasing trend of hypertriglyceridemia has been parallel to that of type 2 diabetes and obesity in the past decades [68]. Very low-density lipoprotein (VLDL) and chylomicrons, which are known as TG-rich lipoproteins (TRLs), are spherical particles with core lipids (TG and cholesterol esters), phospholipids, free cholesterol, and surface apolipoproteins. The origins of TGs are generally exogenous or endogenous. Exogenous TG is mostly obtained from daily diet and transported within chylomicrons, while endogenous TG circulates in VLDL and is mostly formed in the hepatobiliary system. 

The levels of fasting and postprandial TGs depend on the balance between lipoprotein lipase-mediated lipolysis and uptake in the human liver. VLDL overproduction is the most common upstream cause of hypertriglyceridemia, and the inherited capacity of the lipoprotein lipase-mediated lipolysis pathway modulates the steady-state level. Comprehensive evaluations are strongly suggested when hypertriglyceridemia is suspected. Furthermore, insulin resistance, overweight, and type 2 diabetes mellitus may be detected simultaneously in this population. In clinical practice, these conditions are usually treated as metabolic syndrome, which comprises the aforementioned three conditions. Metabolic syndrome may increase VLDL levels, especially when free acids and insulin accumulate in the circulation [70]. An environment with an extremely high free acid concentration, hyperglycemia, and insulin resistance, would result in increased chylomicron secretion, while glucagon-like peptide 1 would play a counterbalancing role in the pathway [70]. Moreover, apoC-III has been shown to decrease the removal of remnants in individuals with high VLDL levels and a higher apoC-III concentration is an important factor leading to dyslipidemia [71].

TG-rich VLDL particles and metabolic remnants are the main transporters of TGs in human circulation. The plasma concentration of TGs has been shown to be parallel to the circulating apo B-containing TRL level, which is known to be associated with ASCVD formation [72]. A non-fasting TG level of 6.6 mmol/L was significantly associated with a 5-fold higher risk of acute coronary syndrome, a 3-fold increased risk of stroke, and a 2-fold increased adjusted risk of all-cause mortality compared to a level of 0.8 mmol/L in population-based cohort studies in Copenhagen [73,74]. These results show the importance of monitoring the TG level in primary ASCVD risk modification. In another study investigating secondary ASCVD risk after acute MI, TGs were found to be significantly associated with both short-term and long-term ASCVD outcomes. Furthermore, most patients in the study had been treated with statins, which further highlights the crucial role of TGs in secondary ASCVD prevention [75]. 

Several studies have used Mendelian randomization and shown that the association between TG concentrations and ASCVD may be causal. Nevertheless, the evidence needs to be interpreted with caution, because nearly all variants associated with TGs were also associated with the trends of HDL-C, LDL-C, and Lp(a) [76,77]. In another study, the authors used Mendelian randomization to show that TG-lowering lipoprotein lipase variants and LDL-C-lowering LDL receptor variants had similar effects on the ASCVD risk per unit change in apo-B [78]. Taken together, these studies demonstrated the causality of TRLs and their remnants on the ASCVD risk, partly due to the plasma level of apo B-containing particles. Another possible mechanism underlying the relationship between TGs and atherosclerosis is the deposition of cholesterol-ester-enriched smaller TRLs on the arterial walls and the subsequent initiation of pro-inflammatory/thrombotic pathways. Furthermore, high circulating TG levels have been associated with pathological HDL-C particles, which could lead to an increased risk of ASCVD [69]. In contrast, the correlation between circulating TG concentrations and ASCVD risk has varied among previous studies and was lost in several multivariate analyses [79]. Moreover, the correlation was reduced after adjusting for non-HDL-C or apoB in an epidemiological study [80].

Collectively, the aforementioned studies demonstrate that TRLs and their remnants play a crucial role in ASCVD risk assessment [81]. According to current clinical guidelines, lowering LDL-C remains the primary treatment goal in the management of dyslipidemia. In addition, clinicians should focus on modifications of TRLs, such as non-HDL-C and apoB, which are highly recommended in the updated guideline [7]. Previous studies on fibrates, niacin, and cholesteryl ester transfer protein inhibitors did not demonstrate a robust or convincing reduction in the risk of ASCVD in an optimal cholesterol-lowering population [82,83]. Nevertheless, several ongoing trials are focusing on the important roles of TRL with respect to the residual ASCVD risk in statin users. The results of these ongoing clinical trials and upcoming evidence regarding omega-3 fatty acids (high-dose icosapent ethyl) [78], and the selective peroxisome proliferator-activated receptor modulator pemafibrate may help to clarify which population will benefit from a reduced risk of ASCVD by lowering TRL levels [84,85]. The development of molecular technologies has provided more detailed information on the pathways underlying TRL modulation. Several emerging therapeutic molecules have been targeted, including inhibitors of angiopoietin-like protein 3 (evinacumab; allele-specific oligonucleotide IONISANGPTL3-LRx) (Table 1) [86], and inhibitors of intestinal diacylglycerol acyltransferase (pradigastat) [87], as well as those targeting apoC-II and A-V and angiopoietin-like protein 4 [88]. TRL modification strategies in specific patients can be expected to become a crucial part of lipid-directed treatment in the near future.

## 6. Apolipoprotein C-III

ApoC-III is a glycoprotein with 79 amino acids that is principally synthesized in the liver and is associated with TRLs [96]. ApoC-III increases the plasma level of TGs by facilitating the assembly and secretion of TRLs in the liver, inhibiting lipoprotein lipase activity, and interrupting the binding of apoE to its hepatic receptors, thus resulting in impaired clearance of TRL remnants [97]. The accumulation of apoC-III on HDL leads to conformation changes of HDL, including decreased apoA-I content, impaired insulin sensitivity, and reduced cholesterol efflux capacity [98,99]. ApoC-III also enhances the expression of adhesion molecules in endothelial cells via the NF-κB pathway and promotes the chemoattraction of monocytes [100]. Besides its role in atherosclerosis, apoC-III has been proposed to induce the apoptosis of insulin-secreting pancreatic β-cells [101].

There is ample evidence showing the strong link between increased apoC-III levels and ASCVD risk [102]. Genome-wide association studies and Mendelian randomization studies have demonstrated a direct relationship between loss-of-function mutations of the *APOC3* gene, lower TG levels, and reduced ASCVD risk [103,104,105,106] Certain genetic polymorphisms have been associated with the upregulation of *APOC3* gene expression and mRNA stability, such the *SstI* gene polymorphism [107]. Other studies have demonstrated a 4% decrease in the risk of ASCVD with every 1 mg/dl decrease in plasma apoC-III level. However, currently available lipid-modifying therapies are not able to target apoC-III level specifically [108,109,110,111].

Antisense oligonucleotide (ASO) targeting the messenger RNA of *APOC3* gene was developed in 2014 to interrupt the transcription of *APOC3* and thus reduce the level of TGs [112]. After showing promising efficacy in reducing the level of TG, apoC-III, and non-HDL cholesterol in phase 1 and phase 2 clinical trials, volanesorsen has been evaluated in two major phase 3 clinical trials. The APPROACH trial was a randomized double-blinded placebo-controlled trial that enrolled 66 patients with familial chylomicronemia syndrome (FCS), a rare genetic disease caused by an inactivating mutation of both alleles of *LPL* genes and genes encoding other proteins required for LPL activity [91]. After three months, volanesorsen resulted in a 76.5% reduction in the plasma level of TGs, equivalent to a reduction of 750–880 mg/dl, and 84.2% reduction in the plasma level of apoC-III. Other effects on the lipid profile included a significant increase in the levels of HDL, apoA1, and LDL, and decreased levels of chylomicron TGs, VLDL-C, and non-HDL-C. However, thrombocytopenia occurred in 76% of the subjects receiving volanesorsen compared to 24% in the placebo group. The COMPASS trial was a randomized, double-blinded, placebo-controlled trial that enrolled 114 patients with severe hypertriglyceridemia (fasting TG ≥ 500 mg/dl) [92]. Similar to the results of the APPROACH trial, the level of TGs was decreased by 73% after three months of volanesorsen treatment. No serious adverse effects regarding platelet count were reported. Another phase 2 randomized double-blinded placebo-controlled trial enrolled patients with hypertriglyceridemia (TG > 200 mg/dL and < 500 mg/dL) and uncontrolled type 2 diabetes mellitus (HbA_1c_ > 7.5%) to test whether Volanesorsen was able to improve insulin sensitivity [113]. In the volanesorsen group, insulin sensitivity improved by 50% and the level of HbA_1c_ decreased by 0.44%. Since insulin resistance is associated with an increased risk of ASCVD, the effect of improved insulin sensitivity and reduced apoC-III may be translated to a reduction in the risk of ASCVD. However, the Food and Drug Administration refused to approve volanesorsen for the treatment of FCS due to safety concerns, and thus a large cardiovascular outcome trial will not be conducted. Structurally, volanesorsen is a 2ʹ-O-(2-methoxyethyl)-modified ASO which has demonstrated good metabolic stability and RNA binding affinity. Despite this modification, less than 15% of the drug is distributed to hepatocytes [114].

A third generation of ASO modified with triantennary N-acetyl galactosamine (GalNAc3) to target the asialoglycoprotein receptors on hepatocytes has been developed. This modification enhances the affinity of ASO to hepatocytes and allows similar efficacy to untargeted ASO with 20–30-fold lower dosing, thus minimizing systemic exposure [115,116,117]. In a phase 1/2a double-blind, randomized, placebo-controlled, dose-escalation study, the novel GalNAc3-modified ASO, AKCEA-ApoCIII-LRx, was administered to healthy volunteers (ages 18–65 years) with elevated TG levels, with predefined doses ranging from 10 mg to 120 mg and varying schedules. After single doses of AKCEA-ApoCIII-LRx of 10, 30, 60, 90, and 120 mg, the median reductions in apoC-III were 0, −42%, −73%, −81%, and −92%, and the mean reductions in TG were −12%, −7%, −42%, −73%, and −77%, respectively. Significant reductions in total cholesterol, apoB, non-HDL-C, VLDL-C, and increases in HDL-C were also noted after multiple doses. The safety of AKCEA-ApoCIII-LRx was demonstrated in this study as no significant effects on the liver or kidney function were noted, and most importantly, no thrombocytopenia events occurred. Recently, a phase 2 trial (NCT03385239) enrolling patients with hypertriglyceridemia and established ASCVD or those at a high risk of ASCVD was completed. It is very interesting to see what effects AKCEA-ApoCIII-LRx may have on the risk of ASCVD.

## 7. Apolipoprotein A-I

ApoA-I contains ten consecutive helical regions, which are critical for its amphipathic properties to facilitate the dissolution of lipids in an aqueous environment [118]. The conformation of apoA-I is highly dynamic and readily accommodates changes in size and composition of HDL particles [119,120,121,122]. After transporting cholesterol to the liver, apoA-I dissociates from HDL particles and continues to initiate de novo RCT and HDL formation. The degree of release of apoA-I from HDL has been associated with the efficiency of cholesterol efflux [123]. Besides promoting RCT, apoA-I inhibits pro-oxidative and pro-inflammatory processes, induces vasodilation, and inhibits the activation of platelets. The cholesterol efflux capacity of apoA-I can be compromised in specific conditions [124]. For example, point mutations of *APOAI* gene may result in decreased HDL lipoprotein formation, stability [125], and lipid-binding affinity [126,127,128,129,130,131,132,133,134]. The A164S point mutation has been shown to result in a series of functional impairments and increased cardiovascular mortality without reducing HDL level [135]. Oxidative modifications of apoA-I by MPO have also been shown to suppress its anti-inflammatory and antiapoptotic activities on endothelial cells in vitro and to activate pro-inflammatory pathways [136]. Certain mutations of the *APOA1* gene may produce MPO-resistant apoA-I forms that are antiatherosclerotic [137].

Previous studies have demonstrated a strong association between low apoA-I levels and increased risk of ASCVD [138,139,140,141]. In a cohort study of subjects suspected of having coronary artery disease who received coronary computed tomography angiography, the serum oxidized HDL/apoA-I ratio was significantly correlated with the prevalence of high-risk plaque and severe luminal narrowing [142]. In patients with type 2 diabetes mellitus, the prevalence and incidence of peripheral artery disease have also been reported to be significantly higher in subjects with low apoA-I levels [143]. On the other hand, a good cholesterol efflux capacity has been shown to promote angiogenesis and the development of collateral circulation in patients with chronic total coronary occlusion [144].

The roles of a well-function apoA-I and HDL have gained increased attention in the past decades as most therapies targeting an increase in HDL concentration have failed to reduce ASCVD risk [48,49,145,146,147,148]. The only CETP inhibitor to demonstrate a beneficial effect is anacetrapib. In the REVEAL trial, patients with ASCVD receiving anacetrapib had fewer coronary events compared to those receiving placebo. However, it took around two years for the full beneficial effects of anacetrapib to emerge, and the rate ratio was only 0.91. In addition, anacetrapib injection was associated with a reduction in non-HDL cholesterol that could be translated into a 10% relative reduction in coronary events. Therefore, it is difficult to independently evaluate the impacts of increased HDL and increased apoA-I on ASCVD risk. 

CSL112 is a plasma-derived apoA-I reconstituted into disk-shaped lipoproteins with phosphatidylcholine and stabilized with sucrose [149]. In the phase 2b AEGIS-I trial, the favorable safety profile of CSL112 infusion was established, as no hepatic or renal adverse effects were noted, and the total and ABCA1-dependent cholesterol efflux increased significantly [94,150,151]. Based on the safety profile, the AEGIS-II study has been designed to assess the efficacy of CSL112 in MACE reduction in high-risk patients with acute MI, and it is currently enrolling subjects. The study protocol has been published elsewhere [152]. In brief, patients with acute MI, multivessel disease, or left main disease, and established risk factors defined as diabetes mellitus or at least two of an age ≥ 65 years, prior history of MI, or peripheral arterial disease will be enrolled. The primary outcome is the time to the first occurrence of a MACE composite of cardiovascular death, MI, or stroke from the time of randomization through 90 days, during which most recurrent cardiovascular events occur. So far, none of the HDL-raising therapies have been demonstrated to decrease the risk of cardiovascular events despite a significant increase in the level of HDL-C by up to 130% or apoA-I by up to 50% [49]. In the AEGIS-I trial, weekly infusions of CSL112 increased the level of apoA-I by nearly 100%. With the results of the AEGIS-II trial, the impact of this potent cholesterol efflux enhancer on the reduction of residual cardiovascular risk may be seen.

## 8. Lipoprotein(a)

Lp(a) is composed of LDL particles and glycoprotein apo(a) covalently bound to the apoB component of LDL [153,154,155]. It is synthesized in the liver and catabolized in the liver and kidneys. In patients with chronic kidney disease, the level of Lp(a) is elevated due to decreased excretion [156,157,158,159] Physiologically, Lp(a) participates in thrombogenesis, wound healing, tissue repair, and vascular remodeling [154,160]. Part of its thrombogenicity comes from apo(a), which is a structural homolog to plasminogen. Apo(a) inhibits the conversion of plasminogen to plasmin and may result in reduced fibrinolysis [161]. Lp(a) affects platelet activation, aggregation, increases the synthesis of plasminogen activator inhibitor-1 and inhibits synthesis of the tissue factor pathway inhibitor [162]. It is also one of the most important carriers of oxidized phospholipid (OxPL), which stimulates inflammatory responses in endothelial cells, and promotes proinflammatory and apoptotic genes expression in macrophages [163,164,165].

The associations between the Lp(a) level and an increased risk of atherosclerosis in different vascular beds, large artery atherosclerotic stroke, and calcification of the aortic valve have been well-demonstrated in large-scale studies [166,167,168,169,170,171,172]. These associations apply to different ethnicities and may be independent of LDL-C. In the Brisighella Heart Study, the Lp(a) level has been shown to predict long-term cardiovascular mortality in the intermediate-risk female and high-risk subjects without a history of ASCVD [173]. Another recently published biobank study showed a linear correlation between the Lp(a) level and incident ASCVD, with a hazard ratio of 1.11 (95% CI, 1.10–1.12) per 50 nmol/L increments [174]. The causal relationship between Lp(a) and the risk of ASCVD has been established by genome-wide association studies and Mendelian randomization studies [175,176,177,178]. Through genetic studies, two single nucleotide polymorphisms, rs10455872 and rs3798220, have been strongly associated with small apo(a) isoforms and high Lp(a) concentration [179,180]. The association between rs10455872 and an increased risk of MI was further demonstrated in a meta-analysis [181]. The combination of rs10455872 and rs3798220 single nucleotide polymorphism in the *LPA* gene have been causally associated with aortic valve stenosis [182]. 

Evidence has consistently shown that Lp(a) contributes to residual ASCVD risk in statin-treated patients. The residual risk of ASCVD has been shown to increase gradually as the Lp(a) concentration rises above 30 mg/dL, and sharply as the concentration rises above 50 mg/dL [166,183,184]. In a recent statement by the National Lipid Association, an Lp(a) level of 50 mg/dl or 100 nmol/L was used as a threshold [185], and an estimated 20% of the world’s population was estimated to have an Lp(a) level above this [186]. According to the 2018 ACC/AHA guidelines, measuring Lp(a) should be considered for subjects with a past or family history of ASCVD [3]. The 2019 ESC/EAS guidelines also recommended measuring Lp(a) to identify subjects with a very high Lp(a) level (≥180mg/dL or ≥ 430mmol/L), especially in those with established ASCVD and recurrent events, familial hypercholesterolemia, premature ASCVD, or a first-degree family member with premature ASCVD [7]. Lp(a)-lowering is a promising method to eliminate residual ASCVD risk [187,188], and an analysis of the ODYSSEY OUTCOMES trial revealed that a 1 mg/dl reduction in Lp(a) by alirocumab was associated with a hazard ratio of 0.994 [189,190]. However, no approved medications can specifically reduce the level of Lp(a).

Statin treatment does not lower but may even increase the level of Lp(a) by 10-20% [191,192,193]. The PCSK9 inhibitors evolocumab and alirocumab have been shown to reduce the level of Lp(a), however, the effects were only modest [194]. Apheresis has been demonstrated to result in a 60 –70% reduction in Lp(a) level and ASCVD risk [195,196,197,198], however, it is reserved for very high-risk patients in a few countries due to its time-consuming process. Recent pharmaceutical research has revealed that subcutaneous injections of ASO against hepatic apo(a) mRNA with PELACARSEN, previously known as ISIS 681257, IONIS APO(a)-LRx, AKCEA-Apo(a)-LRX, or TQJ230, significantly reduced the levels of Lp(a) and OxPL [95,116,199]. In a phase 2 trial, 286 patients with established ASCVD and Lp(a) levels above 60 mg/dl or 150 nmol/L were enrolled to receive PELACARSEN. At 6 months of treatment, the subjects receiving PELACARSEN demonstrated an 80% reduction in the Lp(a) level, compared to a 6% reduction in the placebo group and a 25% reduction with PCSK9 inhibitors. In addition, 98% of the subjects receiving PELACARSEN achieved the prespecified Lp(a) target level of 50 mg/dl or 125 nmol/L. Moreover, PELACARSEN decreased oxidized phospholipids by up to 90%. A favorable safety profile was also demonstrated as no significant change in platelet count, liver function, and renal function were noted. PELACARSEN was structurally modified from its prototype, IONIS-Apo(a), with the addition of a covalently attached GalNAC_3_ complex in order to target the ASGP receptors on the surface of hepatocytes more efficiently [200,201]. The ongoing Lp(a)HORIZON trial (NCT04023552) is a phase 3 randomized placebo-controlled trial that plans to enroll 7,680 subjects with an Lp(a) level above 150 nmol/L and established ASCVD, including prior MI, prior CVA, and significant symptomatic peripheral artery disease to receive PELACARSEN or placebo. The primary outcome is trial committee-confirmed expanded MACEs that include cardiovascular death, non-fatal MI, non-fatal stroke, and urgent coronary re-vascularization requiring hospitalization. The enrollment will be completed by 2024. This trial will provide more essential insights to answer the question of whether targeting Lp(a) reduction can specifically further reduce residual cardiovascular risk [202].

## 9. Conclusions

The novel evidence reported in this study provides deep insights into dyslipidemia, and the changing viewpoint from simply treating the concentration to potentially modifying lipoprotein functions. LDL may undergo structural change by oxidation, glycation, and peroxidation, and these modifications result in progressive atherosclerosis. HDL may undergo post-translational modifications such as oxidation, carbamylation, and glycation, and alteration of its cargo molecules such as SAA, LCAT, apoM, S1P, and PON1.

These modifications could decrease the atheroprotective function of HDL. Currently, no pharmacological approach has been developed to block or reverse these modifications. Research on how these modifications are modulated is still needed and will uncover more potential therapeutic targets. Pharmacotherapies targeting a reduction in TGs, apoC-III, and Lp(a), and an increase in apoA-1, are currently under investigation, and promising preliminary results have been reported. Clinical trials that assess the effects of these therapies on cardiovascular events are underway. These targets may further shape the landscape of dyslipidemia treatment and decrease the residual risk of cardiovascular events.

## Figures and Tables

**Figure 1 biomedicines-09-00985-f001:**
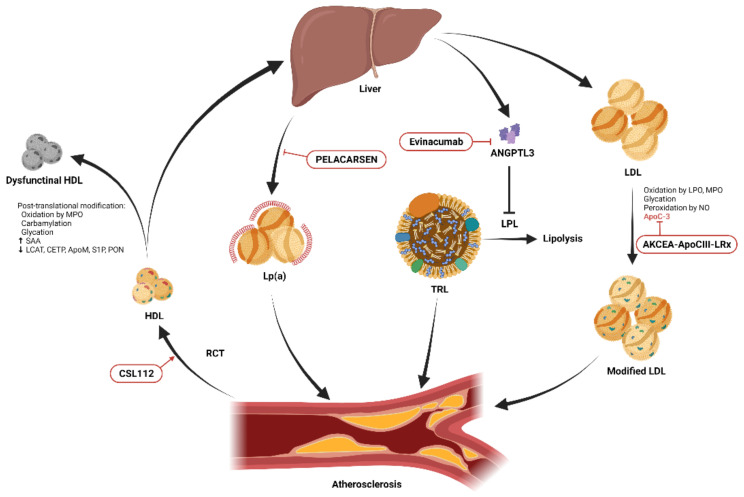
Schematic diagram showing the essential modifications of low-density lipoprotein (LDL) and high-density lipoprotein (HDL) and potential therapeutic targets. HDL promotes cholesterol efflux from cells within atherosclerotic plaques through reverse cholesterol transport (RCT) and transports excess cholesterol from peripheral tissues to the liver for excretion. Post-translational modifications including oxidation, carbamylation, glycation, and alterations of its lipidomic and proteomic structure result in dysfunctional HDL. Infusion of reconstituted apolipoprotein A-I, CSL112, enhances RCT. Lipoprotein(a) (Lp(a)) promotes atherosclerosis through its proinflammatory and antifibrinolytic effects. The production of Lp(a) in the liver can be reduced by the novel antisense oligonucleotide, PELACARSEN. Angiopoietin-like protein 3 (ANGPTL3) produced in the liver inhibits lipoprotein lipase (LPL)-induced lipolysis, resulting in increased circulating triglycerides carried by triglyceride-rich lipoproteins (TRLs), and accelerated atherosclerosis [9]. This process can be blocked by the ANGPTL3 inhibitor, evinacumab. Native LDL is modified by oxidation, glycation, peroxidation, and apolipoprotein C-III (apoC-III) adhesion and becomes more atherogenic. The expression of apoC-III can be suppressed by another novel antisense oligonucleotide, AKCEA-ApoCIII-LRx.

**Table 1 biomedicines-09-00985-t001:** Potential therapeutic targets and emerging pharmacological lipid-lowering approaches.

Potential Therapeutic Target	Pharmacological Approach	Published Clinical Trials	Subjects	Pros	Cons	Ongoing Trials and The Aims of The Trials
Angiopoietin-like protein 3	Evinacumab, a recombinant human monoclonal antibody that inhibits angiopoietin-like protein 3	ELIPSE HoFH (phase 3) [89]	Patients with homozygous familial hypercholesterolemia	47.1% reduction in LDL-C levels Evident LDL-C reduction occurred early after treatment Approval on 11 February 2021 in the USA for use as an adjunct to other LDL-C lowering therapies for the treatment of adult and paediatric patients aged 12 years and older with homozygous familial hypercholesterolemia	Influenza-like illness, pain in extremity, asthenia, constipation, abdominal pain, anaphylaxis High costs, annual cost of the drug estimated to be USD 450,000 on average [90].	NCT03409744To evaluate the long-term safety and efficacy of Evinacumab in patients with homozygous familial hypercholesterolemia NCT04233918To evaluate the efficacy and safety of Evinacumab in pediatric patients with homozygous familial hypercholesterolemia
ApoC-III	Volanesorsen, a 2ʹ-O-(2-methoxyethyl)-modified antisense oligonucleotide	APPROACH (phase 3) [91]	Patients with familial chylomicronemia syndrome	77% decrease in mean TG levels.	25 (76%) in the volanesorsen group had platelet-level decreases to below 140,000 per microliter Decreases in platelet levels were reversible with an interruption in dosing	
		COMPASS (phase 3) [92]	Patients with severe hypertriglyceridemia	73% decrease in TG levels.		
	AKCEA-ApoCIII-LRx, a GalNAc_3_ modified antisense oligonucleotide	Phase 1/2a trial [93]	Healthy volunteers	Dose-dependent reductions of TG levels from -12% to -77% No significant effects on the liver or kidney function and no thrombocytopenia events occurred.		NCT03385239To evaluate the effect of AKCEA-APOCIII-LRx on TG levels in patients with hypertriglyceridemia and established cardiovascular disease NCT04568434To evaluate the effect of AKCEA-APOCIII-LRx on TG levels in patients with familial chylomicronemia syndrome
ApoA-I	CSL112, a plasma-derived reconstituted apoA-I	Phase 2b AEGIS-I trial [94]	Patients with myocardial infarction	A 4.3-fold increase in ABCA1-dependent cholesterol efflux capacity and a 2.45-fold increase in ApoA-I level. No significant change in liver or kidney function		NCT03473223To investigate the effect of CSL112 on major cardiovascular event in subjects with acute coronary syndrome (AEGIS-II)
Apolipoprotein(a)	PELACARSEN, an GalNAC_3_ modified antisense oligonucleotide	Phase 2 trial [95]	Patients with established ASCVD	80% reduction in Lp(a) level Up to 88% decrease in oxidized phospholipids No significant change in platelet count, liver function, and renal function		NCT04023552To assess the impact of Lp(a) lowering with PELACARSEN on major cardiovascular events in patients with cardiovascular disease (Lp(a)HORIZON)

ABCA1, ATP binding cassette subfamily A member 1, ASCVD, atherosclerotic cardiovascular disease; GalNAc_3_, triantennary N-acetyl galactosamine; LDL-C, low-density lipoprotein cholesterol; TG, triglyceride; Lp(a), lipoprotein(a).

## Data Availability

Not applicable.

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
