# Peer review of "Lipids and Lipoproteins in Health and Disease: Focus on Targeting Atherosclerosis"

_biomedicines, 2021, doi:10.3390/biomedicines9080985_

Round 1

Reviewer 1 Report

Dear Editor,

I carefully read the manuscript by Lee et al.

My comments and suggestions for the authors are the following:

  • Line 41: A reference is missing here.
  • Line 46: A reference is missing here.
  • Line 53:  A reference is missing here.
  • Line 76-86: It is not clear why italics were used here. Can authors explain it to me, please?
  • Line 98-99: All the references should be more properly cited at the end of the sentence.
  • Regarding volanesorsen, authors should more appropriately refer to the most comprehensive currently published meta-analysis: PMID: 32458077.
  • The authors should also include in their manuscript information regarding the recently approved bempedoic acid. They should also refer to their effect on creatinine and uric acid serum levels (PMID: 32358698).
  • In general, most of the references are dated and almost obsolete. I suggest the authors to refer to PMID: 27553697, PMID: 33947039, PMID: 32718857 and PMID: 31648549 in their manuscript.
  • Authors should consider to include in the manuscript tables reasuming the main observations.
  • The "Conclusions" should be more deeply discussed in the manuscript.

Author Response

Thank you very much for your questions. We have made the revision accordingly.

  • Line 41: A reference is missing here.
    Response: Thank you for your opinion. Two reference were cited (PMID: 28104770, PMID: 33632204)
  • Line 46: A reference is missing here.
    Response: Thank you for your opinion. Two reference were cited (PMID: 30586774, PMID: 31504418 at line 54 after revision)
  • Line 53:  A reference is missing here.
    Response: Thank you for your opinion. A reference was cited (PMID: 20816951 at line 61 after revision)
  • Line 76-86: It is not clear why italics were used here. Can authors explain it to me, please?
    Response: Thank you for your opinion. This is a mistake. We did not intend to use italics.
  • Line 98-99: All the references should be more properly cited at the end of the sentence.
    Response: Thank you for your opinion. All the references were cited at the end of the sentence after revision (line 107-109 after revision).
  • Regarding volanesorsen, authors should more appropriately refer to the most comprehensive currently published meta-analysis: PMID: 32458077.
    Response: Thank you for your opinion. We cited the review by Fogacci et al. at line 370.
  • The authors should also include in their manuscript information regarding the recently approved bempedoic acid. They should also refer to their effect on creatinine and uric acid serum levels (PMID: 32358698).
    Response: Thank you for your opinion. We included the information regarding bempedoic acid at line 45-48.
  • In general, most of the references are dated and almost obsolete. I suggest the authors to refer to PMID: 27553697, PMID: 33947039, PMID: 32718857 and PMID: 31648549 in their manuscript.
    Response: Thank you for your opinion. The suggested references and relevant content were added. (PMID: 27553697 at line 486-489, PMID: 33947039 at line 208, PMID: 32718857 at line 517, PMID: 31648549 at line 78). Obsolete references were removed, including reference 3, 6, 11, 12, 26, 27, 31, 34, 48, 69, 86, 98, 99, 105, 107, 118, 120, 125, 127, 128, 131, 132, 136, 137, 139, 140, 147, 163, 168, 172, 173, 184, and 196 in the original manuscript.
  • Authors should consider to include in the manuscript tables reasuming the main observations.
    Response: Thank you for your opinion. We made a table 1 to summarize the potential therapeutic targets and emerging pharmacological lipid-lowering approach at page 29.
  • The "Conclusions" should be more deeply discussed in the manuscript.
    Response: Thank you for your opinion. We provided more details in the conclusions section (line 544-549 and line 552-554).

Reviewer 2 Report

The review paper “Lipids and Lipoproteins in Health and Disease: Focus on Targeting Atherosclerosis” by Chih-Kuo Lee et al discusses the evidence on modifications of major lipid and lipoprotein components, including LDL, HDL, TG, apolipoprotein, and lipoprotein(a) and examples of translating of findings from basic research to potential therapeutic targets for drug development. The paper is well-written and documented, and contains a stimulating discussion.  
The authors should provide more details, probably on Tables, regarding the available pharmacological approaches and their pros and cons, and the systematic table dedicates to potential therapeutic targets. 

Author Response

Thank you very much for your questions. We have made the revision accordingly.

The authors should provide more details, probably on Tables, regarding the available pharmacological approaches and their pros and cons, and the systematic table dedicates to potential therapeutic targets. 
Response: Thank you for your opinion. We made a table 1 to summarize the potential therapeutic targets and emerging pharmacological lipid-lowering approach at page 29.
